# Investigating Residual Stresses in Metal-Plastic Composites Stiffening Ribs Formed Using the Single Point Incremental Forming Method

**DOI:** 10.3390/ma15228252

**Published:** 2022-11-21

**Authors:** Andrzej Kubit, Raheem Al-Sabur, Andrzej Gradzik, Kamil Ochał, Ján Slota, Marcin Korzeniowski

**Affiliations:** 1Department of Manufacturing and Production Engineering, Faculty of Mechanical Engineering and Aeronautics, Rzeszow University of Technology, Al. Powst. Warszawy 8, 35-959 Rzeszów, Poland; 2Mechanical Department, Engineering College, University of Basrah, Basrah 61004, Iraq; 3Department of Materials Science, Faculty of Mechanical Engineering and Aeronautics, Rzeszow University of Technology, Al. Powst. Warszawy 8, 35-959 Rzeszów, Poland; 4Institute of Technology and Material Engineering, Faculty of Mechanical Engineering, Technical University of Košice, Mäsiarska 74, 040 01 Košice, Slovakia; 5Department of Metal Forming, Welding and Metrology, Faculty of Mechanical Engineering, Wroclaw University of Science and Technology, Wyb. Wyspianskiego 27, 50-370 Wroclaw, Poland

**Keywords:** residual stresses, SPIF, LITECOR^®^, X-ray diffraction, XRD, single point incremental

## Abstract

Low weight and high strength are significant factors in the current decade’s spread of composite sandwich materials. Previous studies have proven that forming stiffening ribs in these materials through the Single Point Incremental Forming (SPIF) process is possible and gives encouraging results. On the other hand, knowledge of residual stress (RS) values that form during the manufacturing process is essential, as they may affect the structural integrity of manufactured elements, whether in compression or tension. The investigation of the RS in the composite materials formed by the SPIF process using the XRD method was very limited in the previous studies, so this research aims to apply the X-ray diffraction (XRD) method to determine RS on the part of the LITECOR^®^ sandwich material formed using SPIF. LITECOR^®^ consists of a plastic core between two layers of steel. In this study, three types of LITECOR^®^ were used with differing plastic core thicknesses of 0.8, 1.25, and 1.6 mm, while the steel layers’ thickness remained the same at 0.3 mm. The axial and traverse RSs were measured in five positions on both sides of the formed part. It was found that the achieved RSs varied from tensile to compressive along the formed regions. It was found that the residual stress values in both directions were inversely proportional to the thickness of the plastic core. It was noted that the highest RS values were in the unformed base metal, after which the RS was reduced on both sides of the SPIF-formed region, followed by a rise in the RS at the concave of the SPIF-formed region. The maximum measured RS for X-axes was 1041 MPa, whereas, for Y-axes, it was 1260 MPa, both of which were recorded on the back side at a thickness of *t* = 0.8 mm.

## 1. Introduction

Composite materials combine materials with different properties to create a unique material with better features than its component materials [1]. The composite materials industry has achieved remarkable developments and has become the backbone of many industries, especially aviation, where composite materials constitute 50% of the Boeing 787 Dreamliner [2,3]. Many studies have also proven that composite materials can have unexpected uses, such as flame retardants [4]. Composite materials are often used to create high-stiffness and -strength structures and maintain a low weight to keep pace with the increasing global demand for materials whose production requires less energy as a way to face global warming and develop sustainable products.

Regarding composite materials consisting of metals and polymers, the relationship between the metals and polymers is more than just about retaining their advantages; it is a revolution in creating lightweight materials with very high strength [5]. It is essential to mention that in the case of the polymer matrix discretely covering the filler particles, the filler particles are not in direct physical contact, which leads to the containment of the filler chain formation on the close contact surface [6]. A metal-plastic composite sandwich material comprises three layers: upper and lower steel sheets and a plastic core. Density and strength control the cost of the metal–plastic sandwich materials. Aluminum and steel are the primary materials used for most commercial products’ upper and lower layers, while polymers form the core. The main commercial metal–plastic sandwich panels are Alucobond^®^ (Al and core of PE), Bondal^®^ (steel and core of viscoelastic adhesive), Hylite^®^ (Al and core of PP), LITECOR^®^ (steel and core of PE/PA), Steelite^®^ (steel and core of PP), and Usilight^®^ (steel and core of PP, PP/PE) [7]. ThyssenKrupp Steel Europe’s LITECOR^®^ composite material began to appear in several studies, including studies of weldability using resistance spot welding [8,9], mechanical joining based on the mortise-tenon joint [10] and relying on three-stage joining to produce a more significant and stiffer mechanically locked joint [11]. Furthermore, studies expanded to investigate whether a commercial sandwich material could fill the role of automotive and industrial applications [12,13].

Strengthening ribs is a common method for adding strength to a sheet metal component without increasing wall thickness. When the material is disfigured to form the ribs, the material’s effective thickness increases, strengthening the material; this can be a lifesaver and a cost reduction. The instruments for strengthening ribs used punch presses or beaded embosses to decrease the sheet’s thickness without compromising the product’s rigidity and strength. The use of strengthening ribs in composite materials has been studied in the last few years. Song et al. [14,15] confirmed that the modulus of elasticity of carbon fiber-reinforced polymer (CFRP) with strengthening ribs was greater than that of normal CFRP. Davoodi et al. [16] used strengthening ribs to enhance the bumper beam’s performance and impact property by using a hybrid fiber of glass and kenaf as a car bumper beam.

Incremental sheet forming (ISF) is a relatively novel method of shaping plastic that is flexible and has low tooling costs. Instead of the traditional stamping processes, it can be used for one-off production, small series, and prototypes [17,18]. ISF technology is being used more and more to make lightweight structures in the aviation, shipping, and auto industries [19]. It is an excellent way to make parts for prostheses, orthoses, and highly customized medical products [20]. In addition, as the main applications of ISFP are manufacturing unique prototypes in small quantities for testing, ISFP is used for many applications, such as reflective surfaces for headlights in the transportation industry, and ankle support and a part for a knee implant in medical applications, etc. [21].

ISF is distinguished by its capacity to shape workpieces with a CNC milling machine. The possibility of more significant material deformation with decreased susceptibility to cracking is considered an advantage of using the ISF method [22].

The SPIF has been used since 1967, consisting of a backing plate, a blank holder, and a rotating tool placed on a CNC machine for single-point forming on a sheet metal blank, as shown in Figure 1. The main factors in the SPIF are sheet thickness and tool geometry. The tool’s geometry includes step, angle, rotational speed, and transverse velocity [23]. In recent years, Single Point Incremental Forming (SPIF) has been utilized more frequently than any other incremental forming technique (ISF). The principle of the SPIF process in forming the required shapes depends on the generation of specified and controlled small deformations repeatedly and regularly using CNC machines. The majority of SPIF processes are performed using a rounded rigid tool. Occasionally, different tool end shapes were used to get the desired geometry. The speed of the tool’s rotation, the feed rate, the thickness of the sheet, the tool radius, the contact conditions, and the tool path strategy all affect how easily SPIF sheets can be shaped [24]. Recently, SPIF has been used as a tool for creating stiffening ribs to strengthen sheet metal [25,26].

Residual stress measurement is crucial for determining whether a component can withstand the applied stress conditions throughout its service life. In X-ray diffraction (XRD), residual stress is the stress caused by the interaction between the surface of the material being worked on and the SPIF tool.

In SPIF, residual stress can be influenced by machining process parameter adjustments, and the average difference between clamped and unclamped residual stress amplitudes can reach 18% [27]. More than one study showed that it could improve the geometrical precision of drawn pieces by controlling the thickness of the sheets alongside the tool diameter and step-down; when the tool diameter and step-down decrease and the sheet thickness increases, the geometric accuracy improves [28,29]. When there is much deformation in SPIF, the RSs are not evenly distributed. This makes it hard to get the shape and dimensions of the formed parts right.

Several studies attempted to investigate the relationship between residual stress and fatigue to gain a better understanding of their effects on the accuracy of incrementally formed parts. It was found that the effect varies between tensile stresses and pressure stresses and their relationship to fatigue. The compressive RSs enhance the fatigue life of the incrementally formed parts; in contrast, the combination of service stresses with tensile RSs reduces it. In general, RSs have an impact on the fatigue life of protective coatings in the automotive industry, especially for low-carbon steel sheets [30,31].

In most studies during the last decade, the hole drilling method was the primary method in incremental forming processes for measuring the RS. Furthermore, several studies also used unique numerical computations, contours, and slitting methods. Several studies are reviewed here. Radu et al. [32] used the hole-drilling strain gauge method to examine the RS in AA 1050 alloy experimentally and by simulating finite elements. The results indicated that the magnitude of compressive RS varies with sheet thickness. Later, the study was extended by using the strain-rosette method. When small values of vertical step sizes and tool diameter were used, it was possible to obtain a favorable residual stress state and, implicitly, a high degree of part accuracy [33]. Another study focused on AA 1050 alloy was carried out by Abdulrazaq et al. [34], using three tool shape types. It was found that as step size and feed rate values went up, so did RS. Furthermore, Shi et al. [35] focused on Cu/steel-bonded thin composites formed incrementally to determine the most significant parameters on RS when using the hole-drilling method. The study found that the tool radius and wall angle had the most significant effects on RS, while tool rotation had the slightest effect. Hajavifard et al. [36] added RS by turning metastable austenite into martensite with the ISF tool to improve the properties of conical annular discs. Maqbool and Bambach [37] studied the influencing factors of both parallel and perpendicular RSs on the pyramidal frustum formed using SPIF. The results were achieved numerically using ARGUS^®^ software and experimentally using the hole-drilling method. The results of the analyses show that the wall angle was the most crucial factor in how residual stress built up and how accurate the geometry was.

Several studies used the X-ray diffraction (XRD) method to examine the RS. Maaß et al. [38] investigated how residual stress built up in AA 5083 alloy sheets during the SPIF process. In this study, the XRD method was used to measure the RSs in several grooves using a unidirectional tool path; sometimes, a bidirectional strategy was used. The results indicated that the tool path strategy was not a decisive factor in the amplitude of the resulting RS. Tanaka et al. [39] determined the RS through numerical computations where the tool radius and feed rate were the process parameters. The study showed that the residual stress was more effectively affected by the tool radius than by the depth of the feed rate. As the tool radius decreased, the possibility of RS being generated increased. Recently, Slota et al. [40] applied an XRD method to analyze the residual stress formation of truncated cones of steel sheets during the SPIF process. According to the study, the stress profile had a nonlinear distribution, and the highest residual stress was about 84.5 MPa.

Since the investigation of the RS in composite materials formed using the SPIF process using the XRD method was very limited in previous studies, there is a clear research gap in this field. Furthermore, no studies have tried to use the XRD method to figure out the RS in stiffening ribs made using SPIF in metal–plastic composites. Therefore, the motivation of this article is to examine the use of the XRD method in the Corlite^®^ composite material supported by stiffening ribs with different plastic core thicknesses. This study measured the RSs in the tangential and axial directions independently. The stiffening ribs were carried out on a TM-1P vertical milling machine (Hass Automation, Oxnard, CA, USA). In addition, A Proto iXRD Combo (Proto Manufacturing Ltd., Oldcastle, ON, Canada) was utilized as a diffractometer with CrKα radiation to measure the RS.

## 2. Materials and Methods

This study used equipment and tools to form stiffening ribs and investigate the RS in the resulting panel. Figure 2 describes a flow chart of the leading experimental procedures, from supplying the composite panel to using a milling machine with steel pins and lubricant to form stiffening ribs inside the composite panel and, finally, examining the RS. The sections below give the full details of each procedure. The experimental work was done in the Department of Material Science at the Rzeszów University of Technology.

### 2.1. Incremental Forming

LITECOR^®^ composite material (ThyssenKrupp. Essen, Germany) was used in this study. It is a composite sandwich panel consisting of a polymer core (52% PA6, 36% polyethylene) inside two layers of galvanized atom-free steel (CR210IF); a zinc-coated cover is utilized to prevent corrosion, as shown in Figure 3. The mechanical properties of the LITECOR^®^ are shown in Table 1 [26].

There are three cases used in this study based on the thickness of the plastic core in the LITECOR^®^ composites—0.8, 1.25, and 1.6 mm—with the thickness of the steel covers remaining the same in each case at 0.3 mm. A milling machine TM-1P (Haas Automation, Oxnard, CA, USA) was used to prepare the incremental form. A rounded tip pin of 2.5 mm radius HS2-9-2 (1.3348) high-speed steel was used as a forming tool. To reduce the friction of the tool, Mannol SAE 75W-85 (Mannol, Wedel, Germany) was used as a lubricant. The manufacturer’s lubricant properties are as follows: density of 879 kg/m^3^ (at 15 °C), pour point −45 °C, viscosity at 40 °C of 72.4 mm^2^/s, viscosity index 157, and flash point 210 °C.

LITECOR^®^ 100 mm × 160 mm composite panels were used to create the embossing. The formed stiffening ribs had a length of 120 mm and a width of 20 mm, as shown in Figure 4. The depth of embossing D was determined experimentally with a value of 5 mm. A longitudinal groove SPIF tool was used to prepare the required forming matrix. Initially, the shaping was performed using basic parameters selected based on several previous studies [41,42].

A continuous spiral-shaped toolpath was utilized during the forming process. A vertical pitch of 0.4 mm was assumed, along with a feed rate of 1500 mm/min and a rotational speed of 300 rpm.

### 2.2. X-ray Diffraction Analysis

Residual stress measurements were carried out using a Proto iXRD Combo X-ray diffractometer at the Department of Material Science at the Rzeszów University of Technology. In this research, the multiple exposure *sin*^2^*ψ* method and *ω* geometry were used. The *sin*^2^*ψ* method X-ray diffraction (XRD) analysis, introduced in the 1960s, is widely used in measuring RS [43]. Since then, it has been gradually introduced in research and development projects as well as industrial quality control processes for polycrystalline metallic materials. Currently, it can be applied for non-destructive residual stress measurements on the surface of the machine or construction parts. As the compressive RSs in the surface layer are desired, this method is applied for post-process RS measurements in parts subjected to heat treatment (e.g., gears and shafts), welding, surface treatment (e.g., case carburizing, shot peening, grinding), and plastic deformation (e.g., forging, press forming).

The XRD *sin*^2^*ψ* technique was used in this study to measure the RS at the specimen center in both the axial and traverse directions using the setup shown in Figure 5.

There are three cases used in this study based on the thickness of the plastic core in the LITECOR^®^ composites—0.8, 1.25, and 1.6 mm—with the thickness of the steel covers remaining the same in each case at 0.3 mm. For each case, the RSs were measured at five locations for axial (x) and traverse (y) directions for both sides of the SPIF-formed region, as shown in Figure 6.

Residual stress measurements were carried out using a Cr-Kα X-ray tube as a radiation source (wavelength λ_CrKα_ = 0.2291 nm). Crystal lattice strain was determined by the analysis of {211} crystallographic plane diffraction peaks at the Bragg angle 2θ = 156.4°. The following equation was applied to determine strain in the measured direction (Figure 5) [44]:(1)$$\varepsilon_{\Phi \psi }=\frac{1}{2}S_{2}\left(\sigma_{\Phi }\right){sin}^{2}\psi +\frac{1}{2}S_{2}\left(\tau_{\Phi }\right)sin2\psi -S_{1}\left(\sigma_{11}+\sigma_{22}\right)$$
where: *σ_Φ_* is the stress component in the measured direction, *τ_Φ_* the shear stresses, *σ*_11_*σ*_22_ the normal stresses, and 1/2*S*_2_ and *S*_1_ the X-ray elastic constants.

X-ray elastic constants 1/2*S*_2_ = 5.08 × 10^−6^ MPa and −*S*_1_ = 1.27 × 10^−6^ MPa were obtained using Equations (2) and (3) [45].
(2)$$\frac{1}{2}S_{2}=\frac{\nu +1}{E}$$
(3)$$S_{1}=-\frac{\nu }{E}$$

*ψ* angle tilts were used as follows: ±37.00°, ±32.57°, ±27.79°, ±24.00°, ±15.61°, ±13.00°, ±12.00°, ±8.57°, ±8.39°, ±3.79°, and 0.00°. The rest of the measurement conditions were the following: exposure time 2 s, aperture diameter 2 mm, number of exposures per *ψ* angle 15, gain material β-Ti, X-ray tube power 80 W (U = 20 kV, I = 4 mA).

## 3. Results and Discussions

Stiffening ribs in the LITECOR^®^ panel were formed using the Single Point Incremental Forming Method (SPIF) with a TM-1P milling machine with a high-speed steel rounded tip pin and lubricant. Three sandwiches of LITECOR^®^ were used in this study with polymer core thicknesses of 0.8, 1.25, and 1.6 mm.

The obtained RSs for both axes and corresponding positions on the front and back sides are shown in Table 2 and Table 3, respectively.

Figure 7 and Figure 8 show RS variations for the X- and Y-axes for the front and back sides of LITECOR^®^ composites formed using SPIF. The overall RS behavior indicated that the maximum tensile RS was recorded in the unformed region (points 1 and 5), and then the values sloped dramatically on either side of the SPIF-formed areas (points 2 and 4), and finally, the RS was increased at the bottom of the SPIF-formed area (point 3). The achieved back side RS (Figure 7b and Figure 8b) was higher than the corresponding values of the front side (Figure 7a and Figure 8a) for all thicknesses of the polymer core.

Additionally, the RS values found were inversely proportional to the thickness of the polymer core. The highest stress values were found when the polymer core thickness was 0.8 mm. The increase in the RS in the concave region (point 3) in Figure 7 and Figure 8 after its decrease in the adjacent regions (points 2 and 4) during the SPIF process may be due to problems related to the decrease in thickness in this region compared to the adjacent regions, which often occurs when using the SPIF method due to severe plastic deformation [46]. So, during the SPIF process, the resulting values of RS were related to the maximum forming depth and the possibility of cracks arising in the front and back surfaces.

Figure 9 examines the RS at point 3 for different polymer core thicknesses. The results show that the polymer core thickness of 1.6 mm gave the best performance in RS reduction for all directions and sides except the x-axis (front side). Despite this exception, its value was also close to the other thicknesses of the x-axis (front side).

When performing RS measurements, it is important to know the surface structure. In the case of the ribs under consideration, the surface of the shaped sheets was subjected to significant plastic deformation, which influenced the structure of the surface layer. The surfaces of LITECOR^®^ composite covers were galvanized, and the zinc coating was broken in the process of plastic shaping. Figure 10 shows two exemplary areas of the inner side of the crease in which the EDS analysis was performed and also obtained as a result of the spectrum analysis.

Table 4 shows the chemical composition for the indicated points of the surface layer, which indicates that in selected areas, the coating was partially damaged; as a result of friction, zinc galling and local damage to the steel coating occurred. However, most of the surface showed an intact zinc coating.

Local violation of the zinc layer may have been caused by incorrectly selected parameters for the shaping process, as well as insufficient lubrication during the forming of the ribs. Further tests are required to demonstrate the possibility of plastic forming using the incremental sheet forming method without damaging the protective coating of the steel.

Figure 11 shows SEM micrographs of the inner surface of the rib at different magnifications. Linear traces of deformation caused by the movement of the shaping tool are visible on this surface. As can be seen at higher magnification, the phenomenon of breaking and overlapping of the zinc coating layer fragments took place here as well.

The orange peel phenomenon was observed in selected areas of the greatest plastic deformation of the cover sheets on the outer surface of the stiffening rib, as shown in Figure 12.

The orange peel phenomenon was observed at the ends of the stiffening ribs in the middle of their height, and this is how the plastic deformation of the composite facings took place most often. This is confirmed by the cross-section view of the embossing presented in Figure 13.

## 4. Conclusions

This article used the XRD method to measure the RS in the composite materials formed using the SPIF process. The 2D residual stress distribution along the front and back sides was examined. The most significant RS was found in the unformed zone, after which the values substantially sloped on either side of the SPIF-formed portions, and eventually, the RS increased at the bottom of the SPIF-formed area. For all polymer core thicknesses, the achieved rear-side RS values were more significant than the front-side values. The outcomes demonstrated that residual stress reduction performance was optimum with a polymer core thickness of 1.6 mm.

Based on the analysis of the properties of the rib surface, it was found that, due to plastic deformation and friction between the tool and the inner surface of the shaped material, the zinc coating was damaged locally, but the overwhelming majority of the surface retained the correct zinc coating. Local violation of the galvanization may have been caused by incorrectly selected parameters for the shaping process, as well as insufficient lubrication. Further tests are required to demonstrate the possibility of plastic forming using the incremental sheet forming method without damaging the protective coating of the steel.

## Figures and Tables

**Figure 1 materials-15-08252-f001:**
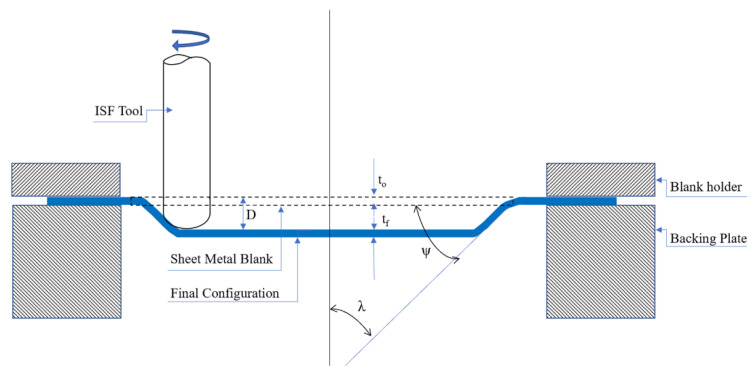
Symmetric SPIF Schematic representation.

**Figure 2 materials-15-08252-f002:**
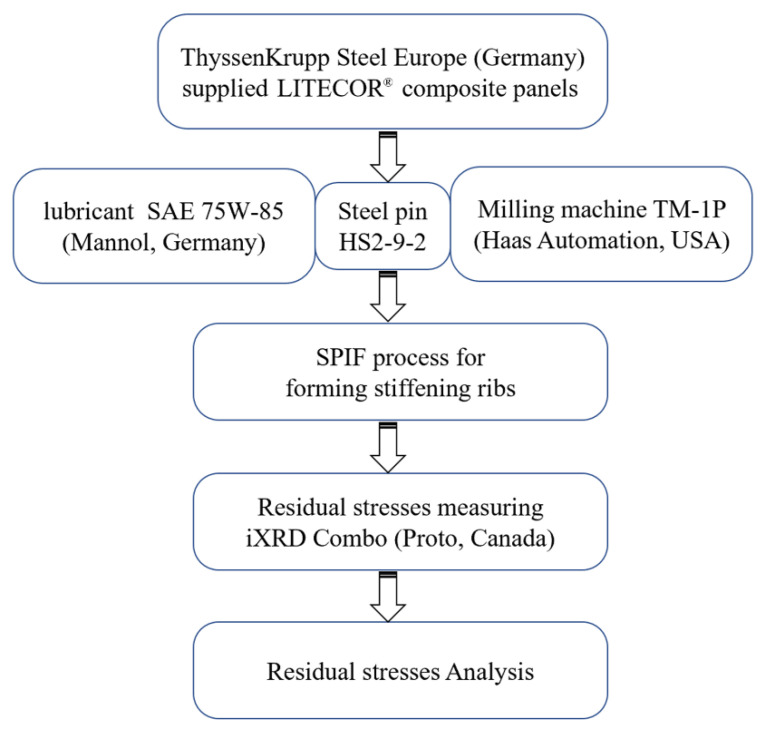
Flow chart of the experimental procedures.

**Figure 3 materials-15-08252-f003:**
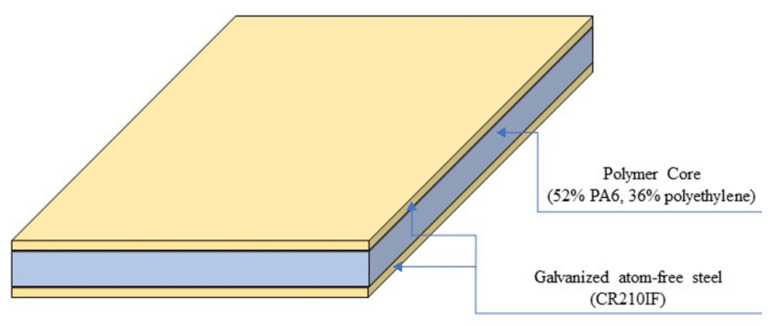
Normal LITECOR^®^ layers.

**Figure 4 materials-15-08252-f004:**
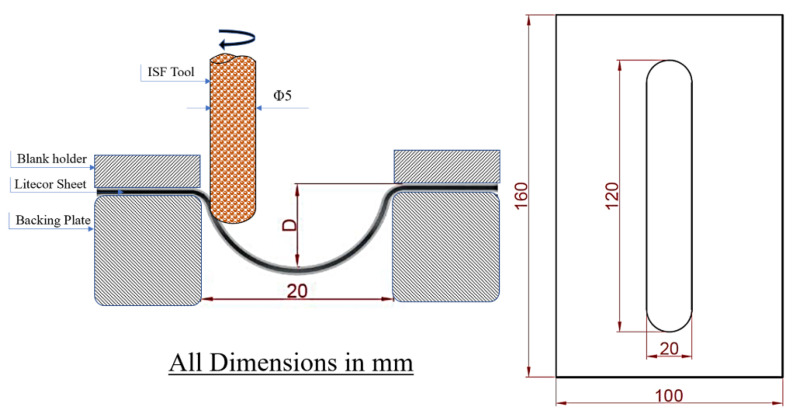
SPIF tool scheme and dimensions of the panel with stiffening rib in LITECOR^®^.

**Figure 5 materials-15-08252-f005:**
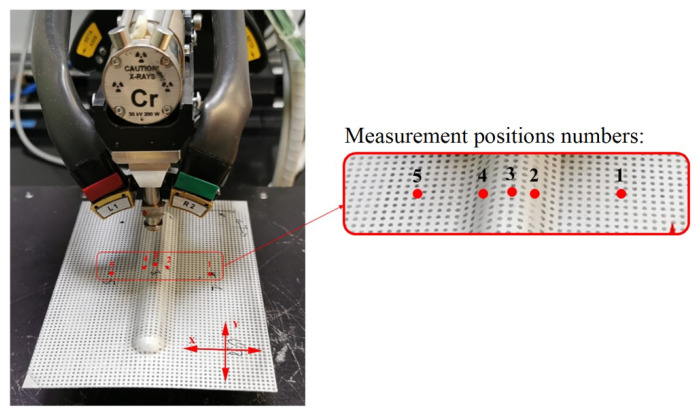
XRD equipment and residual stress corresponding positions 1–5 in LITECOR^®^ panel.

**Figure 6 materials-15-08252-f006:**
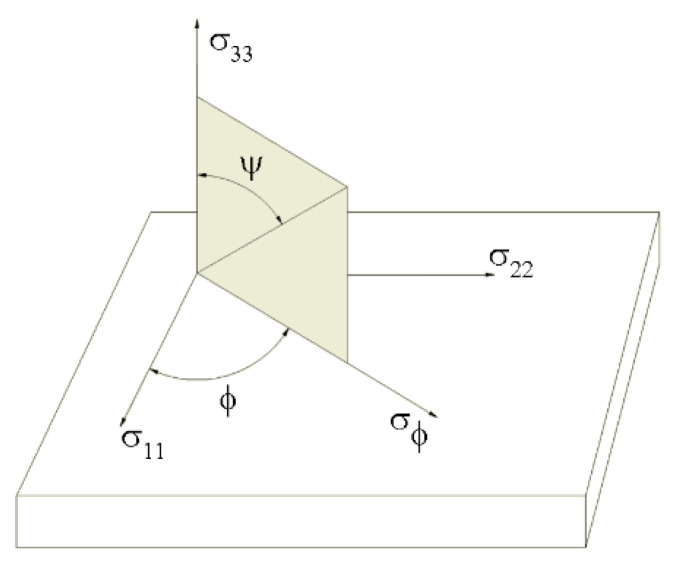
Plane-stress elastic model: *σ*_11_, *σ*_22_, *σ*_33_—normal stresses, *σ_Φ_*—stress component in measured direction, *ψ*—angle of diffracting crystallographic planes.

**Figure 7 materials-15-08252-f007:**
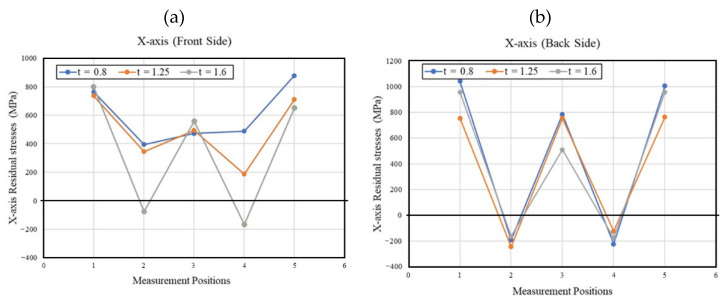
Residual stresses with corresponding measurement locations for the X-axis of the (**a**) front side, (**b**) back side.

**Figure 8 materials-15-08252-f008:**
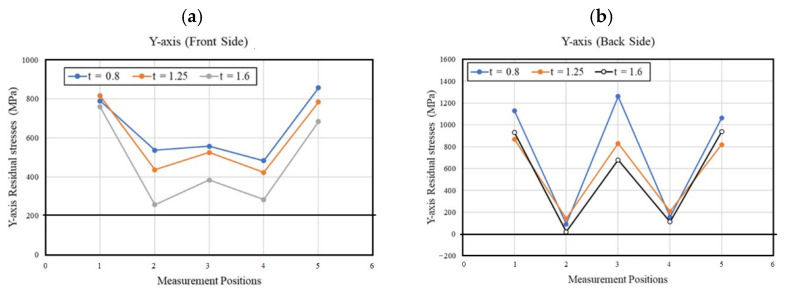
Residual stresses with corresponding measurement locations for the Y-axis of the (**a**) front side, (**b**) back side.

**Figure 9 materials-15-08252-f009:**
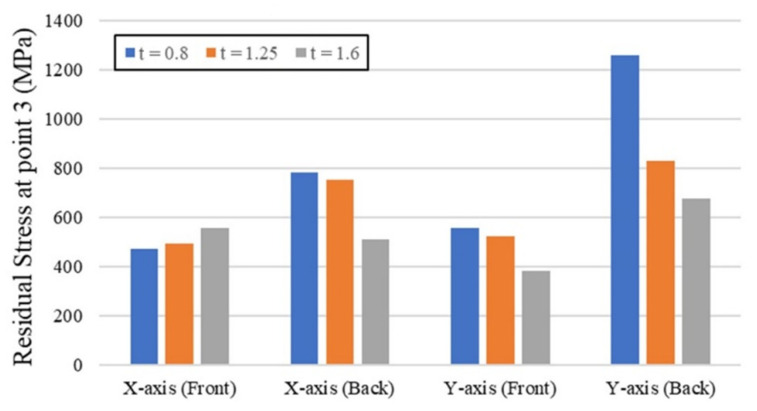
Residual stresses at point 3 for different polymer core thicknesses.

**Figure 10 materials-15-08252-f010:**
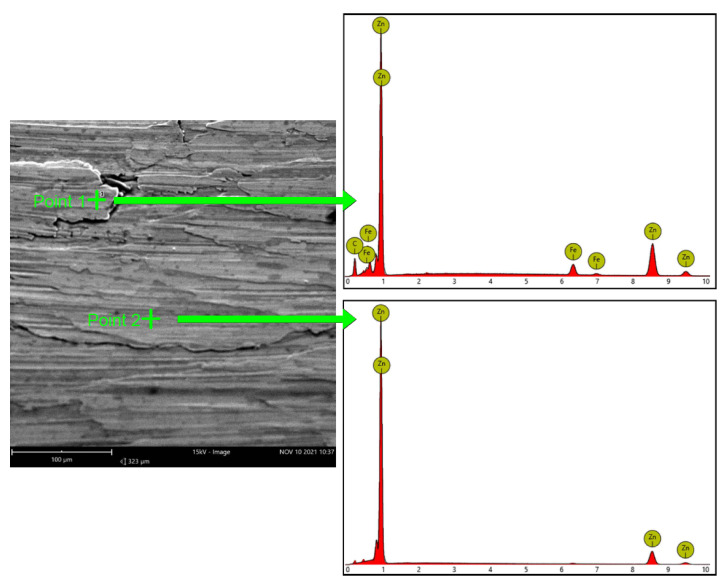
Results of energy-dispersive X-ray spectroscopy (EDS) analysis for the front (inner) surface of the stiffening rib.

**Figure 11 materials-15-08252-f011:**
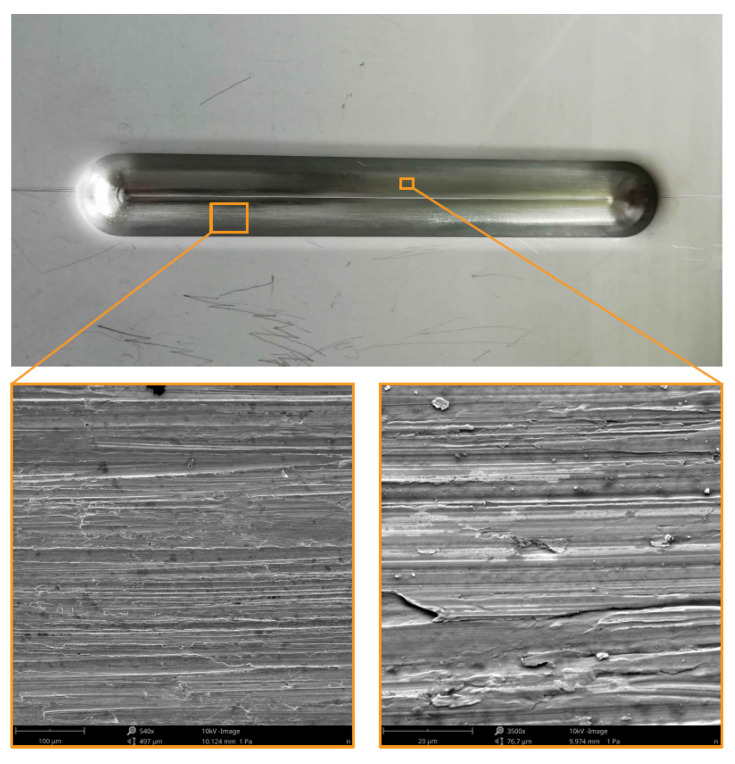
Macroscopic view and SEM images of the inner surface of the stiffening rib.

**Figure 12 materials-15-08252-f012:**
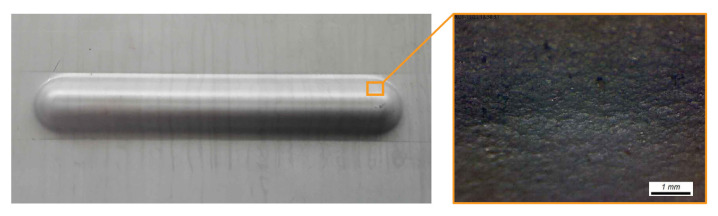
View of the orange peel phenomenon on the outer surface of the stiffening rib.

**Figure 13 materials-15-08252-f013:**
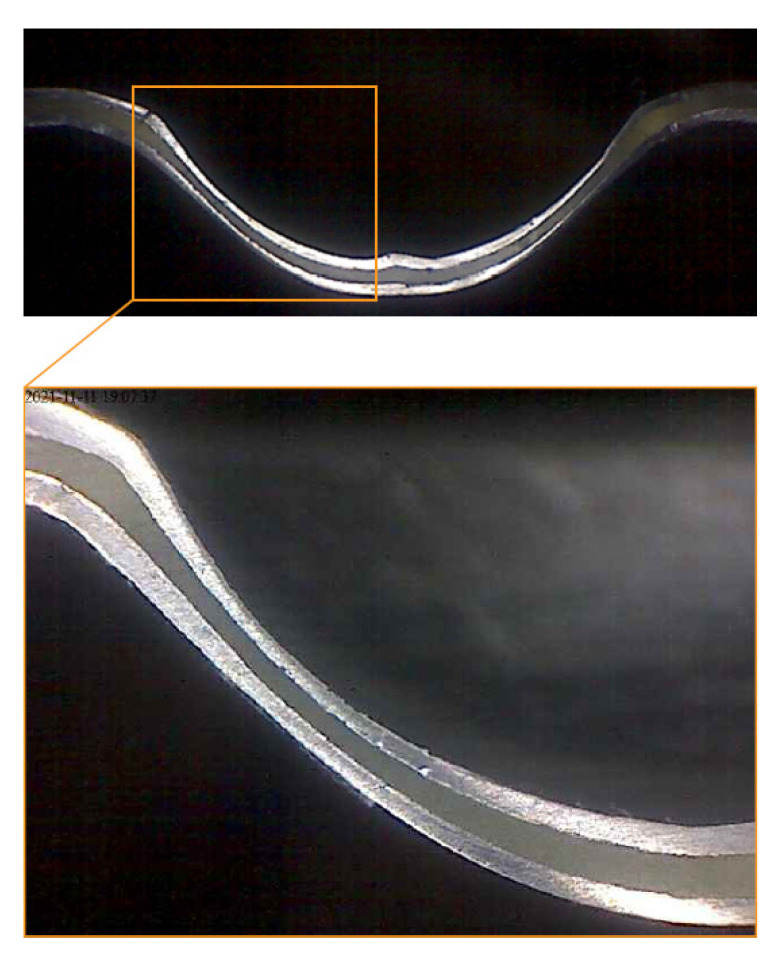
Cross-sections of the end of the stiffening rib.

**Table 1 materials-15-08252-t001:** LITECOR^®^ mechanical properties [26].

Properties	Unit	Value
Yield Strength	MPa	120–180
Ultimate Tensile Strength	MPa	190–240
Elongation	%	28

**Table 2 materials-15-08252-t002:** Residual stresses for the X and Y axes of the LITECOR^®^ front side.

Location	Front Side Residual Stresses for X-Axes (MPa)			Front Side Residual Stresses for Y-Axes (MPa)		
	*t* = 0.8	*t* = 1.25	*t* = 1.6	*t* = 0.8	*t* = 1.25	*t* = 1.6
1	765	736	801	790	816	759
2	395	345	−75	536	436	257
3	473	494	557	557	525	384
4	488	187	−166	483	422	283
5	879	711	651	857	784	684
Max	879	736	801	857	816	759

**Table 3 materials-15-08252-t003:** Residual stresses for the X and Y axes of the LITECOR^®^ back side.

Location	Back Side Residual Stresses for X-Axes (MPa)			Back Side Residual Stresses for Y-Axes (MPa)		
	*t* = 0.8	*t* = 1.25	*t* = 1.6	*t* = 0.8	*t* = 1.25	*t* = 1.6
1	1041	755	956	1130	869	932
2	−193	−246	−164	92	141	21
3	784	753	509	1260	830	678
4	−224	−124	−176	158	208	112
5	1004	762	957	1062	819	937
Max	1041	762	957	1260	869	937

**Table 4 materials-15-08252-t004:** Chemical composition (at%) of the EDS points.

Point Analyzed	Percentage Weight Concentration (%)	
	Zn	Fe
Point 1	87.91	12.09
Point 2	100	-

## Data Availability

Not applicable.

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
