# Peer review of "Investigating Residual Stresses in Metal-Plastic Composites Stiffening Ribs Formed Using the Single Point Incremental Forming Method"

_materials, 2022, doi:10.3390/ma15228252_

Round 1

Reviewer 1 Report

This paper investigates the investigating residual stresses in metal-plastic composites stiffening ribs formed by the single point incremental forming method. This topic is interesting. However, it needs some room of improvement to enhance the quality of the paper. Therefore, I recommend the minor revision before it can be considered to be published.

Several aspects need to be improved:

1.     The abstract is well written with proper introduction, objective of study, methodology and general results of the research. However, it could be more critical if the authors add discussions on the motivation of research with research gap before the research objective. Please add and summarized all the main findings with a comprehensive argument which indicate the significant of the results. The abstract could be more analytical in term of discussion if the authors clearly define the numerical values of the findings such as residual stresses in metal-plastic composites stiffening ribs in order to show the significant of results to readers. Usually, a good abstract writing required to be closed with general conclusion and potential applications to illustrate the general idea of end product for the research.

2.     The introduction section seems quite short to explain the literature survey of the current research work which only referred to 33 references. It should be noted that a good Introduction section should elaborate with comprehensive background literature survey by citing previous work to bridge the scientific gaps from established works. This should have given justification for the present study, which should be followed by the objectives of this study. From the available literatures, it can be found that there are various works on the metal-plastic composites. This would have led to justification for the present study. Therefore, it is suggested to start the first paragraph with general idea on metal-plastic composites as stiffening ribs which bring to second paragraph which discuss on improvement work on the residual stresses. In this paragraph, it should be discussed on the type of modification of metal-plastic composites and how it works toward the structural applications. Recent literatures findings should be added to show the quality of your literature survey. For third paragraph, the authors should introduce the usage of stiffening ribs in metal-plastic composites and how it does enhance the metal-plastic composites toward desired usage. The fourth paragraph should be explained the research motivation, research gaps and problem statement to establish the research objective. At the end, potential applications of this research should be included to show the significant of your work. Relevant article on additives in composites for structural applications should be cited as 10.3390/polym13111701.

3.     For section 2, it is a good writing on methodology section. However, please add on a figure regarding the overall flow chart on the methodology part from procurement of materials up until characterization. Additionally, for each test which note that the machine should be stated its model, brand, where the location of manufacturer and manufacturers name. Also, please indicate to the reader where the tests were performed specifically (laboratory name).

4.     For results and discussions, the discussion should relate the experimental findings with other similar previous literatures. Please correlate them in order to establish a comprehensive discussion and analysis. Please do make a summarization table to indicate the all outcomes from the experimental work.

5.     For conclusion part, it is do not reflect what had been achieved including many speculations. It is too long and should be in one paragraph. Hence these need to be suitably modified. It may be remembered that this Section forms a summary of all the major observations/ results obtained. Accordingly, here presentation should consist of the main Results or the observations of the study briefly. Moreover, the authors have to include and highlight also the objectives and novelty of the work. Please indicate the future work may be conducted in this section. Hence better to rewrite this Section based on the comments given in the whole text.

6.     Throughout this paper, there is need for better language throughout the manuscript.

7.     Generally, the paper though contains some interesting results and novelty work, it lacks in its proper presentation in the whole manuscript. In view of these, the paper is highly recommended and should be accepted for publication in the revised form. It is suggested that the authors should revise the paper in the light of above comments/suggestions.

Reviewer 2 Report

  1. SPIF process needs to explain if possible SPIF process figure can be added in the introduction.
  2. The author needs to check the citation in line 50.
  3. In the introduction, the author must include the application of SPIF-processed materials.
  4. Figure 3 in the manuscript is taken from somewhere or its author's own. If the figure is taken from somewhere proper citation is required.
  5. The method of citation of reference in lines 186 to 190is not appropriate.
  6. In this paper, the author has only varied the sandwich material thickness and measured the residual stress and no information about other process parameters not discussed whether they are variable or constant.
  7. The result discussion is very weak and causes of variation of residual stress are not firmly mentioned. The author discussed only possibilities of variation. It is suggested to conduct SEM on the deformed surface for better understanding.

Reviewer 3 Report

Using the X-ray diffraction method to determine the residual stress on the part of the LITECOR® sandwich material formed by Single Point Incremental Forming is an exciting approach. Reducing weight while increasing strength has always been a key factor for workpieces, including when using sandwich materials. The influence of stiffening ribs through the single-point incremental forming process on the residual stress values after the manufacturing process is of essential importance. The submitted manuscript was an easy read, although I have the following comments as a reader.

50: Reference [6] is given twice.

110-131: You could redesign this paragraph. Study targets, paragraphs 113 and 126, are in context to well-known knowledge [29], line 119. In addition, the result is already announced when the research goal is described, line 117.

154: Here, you could enter the value of D.

161: For better readability, it should be stated in the text for Figure 2 that the length is in mm.

177: In Figure 3, you can insert numbers 1-5 after "..position".

186 and 196: Why are no numbers given here for the literature references but the full text?

240: The figure has the number 7.

Round 2

Reviewer 2 Report

Moderate English changes required